# Role of Cachexia and Fragility in the Patient Candidate for Cardiac Surgery

**DOI:** 10.3390/nu13020517

**Published:** 2021-02-05

**Authors:** Calogera Pisano, Daniele Polisano, Carmela Rita Balistreri, Claudia Altieri, Paolo Nardi, Fabio Bertoldo, Daniele Trombetti, Laura Asta, Maria Sabrina Ferrante, Dario Buioni, Calogero Foti, Giovanni Ruvolo

**Affiliations:** 1Department of Cardiac Surgery, Tor Vergata University Hospital, 00133 Rome, Italy; claudia.altieri@ptvonline.it (C.A.); pa.nardi4@libero.it (P.N.); fabio.bertoldo@uniroma2.it (F.B.); daniele.trombetti@famigliatrombetti.it (D.T.); astalaura92@gmail.com (L.A.); fesab882011@libero.it (M.S.F.); docyuk@libero.it (D.B.); giovanni.ruvolo@uniroma2.it (G.R.); 2Physical and Rehabilitation Medicine, Tor Vergata University of Rome, 00133 Rome, Italy; daniele.polisano94@gmail.com (D.P.); foti@med.uniroma2.it (C.F.); 3Department of Biomedicine, Neuroscience and Advanced Diagnostics (Bi.N.D.), University of Palermo, 90133 Palermo, Italy; carmelarita.balistreri@unipa.it

**Keywords:** frailty, vascular aging, age related syndrome, sarcopenia, malnutrition

## Abstract

Frailty is the major expression of accelerated aging and describes a decreased resistance to stressors, and consequently an increased vulnerability to additional diseases in elderly people. The vascular aging related to frail phenotype reflects the high susceptibility for cardiovascular diseases and negative postoperative outcomes after cardiac surgery. Sarcopenia can be considered a biological substrate of physical frailty. Malnutrition and physical inactivity play a key role in the pathogenesis of sarcopenia. We searched on Medline (PubMed) and Scopus for relevant literature published over the last 10 years and analyzed the strong correlation between frailty, sarcopenia and cardiovascular diseases in elderly patient. In our opinion, a right food intake and moderate intensity resistance exercise are mandatory in order to better prepare patients undergoing cardiac operation.

## 1. Introduction

The concept of frailty was first evidenced in the 1979 [1] and entered in the common medical language, thanks to recognized value in predicting the risk to many chronic diseases in old population, evidencing the marked differences in the two sexes (especially in female people), with respect to the traditional risk factors for these diseases, and in facilitating (or precisely quantifying) the increase of health age-related deficits. Nevertheless, its definition remains uncertain, although three researchers have advanced some major proposals: (1) Fried [2] defines frailty as the process that decreases the physiological reserves and results in a major vulnerability to stressors (pathologies, surgery); (2) Rockwood [3] describes it as the result of the presence of adverse variables in old people, including those of cardiovascular nature (i.e., hypertension, heart attack and arrhythmia); (3) Gobbens [4] suggests that damages in the psychosocial sphere of an individual cause some adverse effects to the health. Currently, advances in the field propose frailty as major phenotype of accelerated aging characterized by a multiorgan dysfunction and/or significantly associated with an increased vulnerability to diverse diseases (multimorbidity) in elderly people [5]. Sarcopenia can be considered a biological substrate of physical frailty [6]. Muscle loss typically begins in the fifth decade of life and proceeds at a rate of decline of 0.8% years [7]. Epidemiological data suggest a wide variability in the prevalence of sarcopenia, depending on the type of population studied, sex, age and diagnostic criteria used. The prevalence of sarcopenia is between 7.5% and 77.6% [8]. There are numerous factors responsible for this muscle loss: the aging process, genetic susceptibility, environmental factors, such as suboptimal diet, prolonged bed rest, sedentary lifestyle, chronic diseases and drugs [9,10]. In most cases the etiology of sarcopenia is multifactorial and sarcopenia is considered PRIMARY (age-related) when the only obvious cause is aging [11,12,13,14]. Malnutrition plays a key role in the pathogenesis of sarcopenia and frailty. The malnutrition refers to an imbalance condition of protein or other nutrient imbalance, responsible for negative effects on body composition, physical function and clinical outcome [15]. Although, malnutrition is not inevitably associated with the aging process. Numerous causes can contribute to a decline in nutritional status: anorexia, edentulism, dysgeusia, dysphagia, motor and visual disability represent physiological and physical causes that can compromise an adequate intake of nutrients [16]. We will see in this narrative review the correlation between frailty, sarcopenia and malnutrition in the management of the elderly patient. At the same time, we proposed a right food intake in order to better prepare patients undergoing cardiac surgery.

## 2. Materials and Methods

### 2.1. Data Sources and Search Strategy

Current literature investigating frailty, sarcopenia and malnutrition is analyzed and contextualized in this review. Specifically, research was conducted on Medline (Pubmed) and Scopus. To review recent studies on frailty, sarcopenia, malnutrition, and cardiovascular disease, we selected scientific papers published in English 10 years since the European Working Group on Sarcopenia in Older People (EWGSOP) was published in 2010 [17]. We used the search term frailty, sarcopenia, malnutrition, cachexia, cardiovascular disease, mortality and morbidity, cardiac surgery.

### 2.2. Study Selection

#### 2.2.1. Inclusion Criteria

The inclusion criteria for the included studies in this review were as follows: (1) assessment of frailty and sarcopenic patients; (2) inclusion of both gender and all races; (3) examination of the impact of undernutrition, sarcopenia and frailty on clinical outcomes; (4) frailty evaluation; (5) evaluation of muscle strength and/or muscle mass for diagnosing sarcopenia; (6) evaluation of the correlation between frailty/sarcopenia and cardiovascular diseases; (7) identification of frailty biomarkers in predicting vascular aging and cardiovascular disease; (8) morbidity and mortality in frailty patients underwent cardiac surgery; (9) application of a specific dietary intake in order to prevent sarcopenia in cardiac surgery patients.

#### 2.2.2. Exclusion Criteria

Editorial, case report, letters to editor, and conference abstracts were excluded from this review.

## 3. Frailty Definition and Quantification

Two main models have been proposed for the frailty evaluation: the phenotypic (primary frailty) or the deficits accumulation model (secondary frailty). Different instruments have been proposed for measuring frailty. Of note are the data from the Cardiovascular Health Study [18] that evidenced in about 25% of older participants signs of frailty without either multiple comorbidities or disabilities (physiological ageing). In this context, frailty has been defined as “primary frailty”, with a phenotypic presentation involving the decline in physical functions and psychological status, without taking into consideration associated diseases or pathological conditions. In measuring the primary frailty, the Fried’s phenotype frailty index has been widely adopted [2]. It derived from an analysis of five health factors: nutrition; physical exhaustion; low energy expenditure (or inactive status); mobility and muscular strength. Deterioration in one of these examined factors was scored as 1 if present or 0 if absent, giving a potential score spanning from 0 to 5. The phenotypic model permitted to classify three groups of individuals: robust (no deterioration); pre-frail (one or two altered factors); or frail (three or more altered factors). This classification was independently correlated with outcomes, such as survival, falls, disability, and institutionalization. The secondary frailty considered the accumulation of multiple deficits, including symptoms, signs, disabilities, pathological conditions, and abnormal laboratory values. Furthermore, its evaluation was based on frailty index based on the defects accumulation’s model [19]. Every deficit has been coded as binary (1 or 0) or ordinal (0, 0.5, 1), consequently the frailty index was the sum of the deficit’s values divided by the total number of deficits listed. This approach evidenced an important issue on measurement of frailty based on phenotypic or deficits accumulation model, that has revealed it is complex and time consuming. Alternative and easier instruments have been, subsequently, proposed for fragility assessment both in general population or in clinical practice [20]. For frailty estimation in general population, two simple scales and multifaceted tools requiring comprehensive geriatric assessment (CGA) have been introduced. Among the scales, the most important tool adopted is the Edmonton frailty scale (EFS) [21]. It examines nine domains of frailty (cognition, general health status, functional independence, social support, medication usage, nutrition, mood, continence, functional performance). The results have been reported in a scale ranging from 0 to 17 values and the participants have been conventionally classified into three categories. A higher score has been associated with a higher degree of frailty. The same nine domains of frailty have been also assessed by using a specific tool requiring comprehensive geriatric assessment. For example, this is the case of the Mini-Mental State Examination (MMSE) [22] or the Geriatric Depression Scale (GDS) [23,24]. In contrast, the functional performance has been detected by using the Handgrip strength test [25,26]: handgrip measurement is assessed on the dominant hand using a Jamar dynamometer adhering to the standardized protocol recommended by the American Society of Hand Therapists and the average value of the handgrip in the two genders is used to define the scores. Thus, a lower score than 30 kg for man and lower than 20 kg for women is considered weak [27]. The valuation of the nutritional status has been performed using the Mini Nutritional Assessment (MNA) [28]. The MNA is composed of 18 items divided in four categories: anthropometric assessment, general state, dietary assessment, and self-assessment. A score ≥24 points indicates a good nutritional status. A score from 17 to 23.5 points is an indicator of a risk of malnutrition, while a score ≤17 points indicates malnutrition. In the appraisal of the general health status, the assessment of the sarcopenia [29], osteoporosis [30], and serum albumin [31] is important. Beside the instruments used to estimate fragility in general population, it is necessary to mention the main tests used in hospital environment. Among these the *SHARE-FI scale* [32] (Survey on Health, Ageing and Retirement in Europe Frailty Instrument) is one of the most counted. This instrument is based on the first wave of the Survey of Health, Ageing and Retirement in Europe, a large population-based survey (*n* = 31,115) conducted in 2004–2005 in 12 European countries. It measures five variables approximating Fried’s frailty definition (exhaustion, weight loss, weakness, slowness, and low activity) with different coefficients for men and women. Another important scale is the SPPB [33] (Short Physical Performance Battery). This battery analyzes physical performance features as 4-m gait speed, balance capacity and sit-to-stand time. The lower extremity performance is a long-term mortality predictor independent of NYHA class and ejection fraction in elderly hospitalized patients. The EFT [34] (Essential Frailty Instrument) is a specific scale used in cardiac surgery. This scale includes biological, physical and mental state thus it may identify subclinical frailty; in fact, its features are albumin and hemoglobin blood values, sit-to-stand time and MMSE test. In the context of cardiac surgery two important risk scores have been introduced in order to predict 30-day mortality and 1-year mortality: respectively, the Comprehensive assessment of frailty [35] (CAF) and the Frailty predicts death One years after Elective Cardiac Surgery Test (FORECAST) [36]. The CAF is composed of different items to quantify the physical performance and coordinative abilities of the patients in addition to scores that are already used to define frailty in medicine. In addition, several laboratory tests result as creatinine and FEV_1_. The FORECAST is a simple version of the CAF with a higher predictive power. It is composed of those five test items (chair rise, weak, stair, clinical frailty scale from the Canadian Study of Health and Aging, serum creatinine level).

## 4. Sarcopenia as Biological Substrate of Physical Frailty

Sarcopenia is considered the biological substrate of physical frailty. The prevalence of sarcopenia is higher in male with low body max index (BMI) [37,38]. Sarcopenia is a common condition in the elderly but can also be seen in younger patients. It is defined *primary* or age related when no cause is highlighted, other than the aging process. It is considered *secondary* when one or more causes are identifiable and in this case it is called activity-related, disease-related, nutrition-related [39]. Sarcopenia is a syndrome characterized by the progressive and generalized loss of mass, muscle strength, physical performance, which leads to an increased risk of disability, poor quality of life, falls, numerous complications, and death [40]. Muscle trophism is a consequence of a balance between anabolic triggers (insulin, physical exercise, amino acids, adrenaline, testosterone) and catabolic triggers (cortisol, catecholamines, glucagon, cytokines, intense exercise) [41]. In the elderly, the catabolic state is associated with the normal aging process, which becomes predominant when particular conditions of comorbidity are concomitant. In these cases, the muscle mass suffers the effects of the general catabolic state in which the body is found. Several factors contribute to the pathophysiology of sarcopenia [42]. In particular the main factor are reduction of sex hormone levels, reduction of growth hormone levels, increased production of cytokines, interleukin-1 (IL-1), interleukin-6 (IL-6), Tumor Necrosis Factor-Alpha (TNF-alpha), alteration of the cellular redox-status, neuromuscular changes, physical inactivity, and malnutrition [43,44]. Drugs can also play a protective or causative role in the development of sarcopenia. Statins, sulfonylureas, glinides have a potential harmful effect on muscle metabolism; while ACE inhibitors, allopurinol, Vitamin D play a protective role on muscle function [45,46]. The muscle is formed of different types of muscle fibers: slow fibers (type I) and fast fibers (types IIa and IIb). With aging, especially in sarcopenic patients, there is a reduction in the diameter of muscle fibers, as well as a progressive loss of fast fibers which results in a reduction in strength, coordination of movements, and walking speed. This happens because the lost fast muscle fibers are replaced by slow fibers. Given the dynamic nature of neuromuscular remodeling, it has been seen that the muscle of the elderly subject under certain triggers maintains the ability to respond and to adapt to the new state. It has been shown that even lifestyle alone can greatly influence the development of muscle mass. This means that effective therapeutic intervention could be applied in order to reverse the processes that lead to sarcopenia [47,48].

## 5. Sarcopenia Diagnosis

The simultaneous presence of muscle mass loss associated with reduced muscle strength or physical performance is recommended for the diagnosis of sarcopenia. There are various methods for the assessment of the sarcopenia.

The muscle mass usually is calculated using the Impedancemetry [49]. This is a valid and recognized alternative to more complex and expensive methods, such as magnetic resonance imaging (MRI), dual X-ray absorptiometry (DEXA) and computed tomography (CT). The exam lasts a few minutes, is absolutely painless, safe and allows you to know the body areas in terms of fat, lean tissue and water content. It is based on the principle that tissues full of water and electrolytes offer less resistance to the passage of an electric current than adipose tissue. The result is then compared with the reference values obtained according to normalization formulas for race, age, sex, body weight. Muscle strength is measured through the Handgrip (dynamometer), a simple tool that evaluates the force developed by gripping the hand; usually three tests are performed, of which the best is chosen [50]. The result is compared with threshold values calculated according to age, sex and BMI. Physical performance can be assessed through the quick and easy walking speed test [51]. The main symptoms related to sarcopenia are muscle weakness and fatigue. This concept does not only concern the bedridden people but also the person who has functional autonomy. This condition does not only concern the thin and undernourished patient, but even obese patient with increased body max index that have a reduction in muscle mass. This scenario is called sarcopenic obesity [52]. Sarcopenic obesity is related to an increased cardiovascular risk, due to the unfavorable metabolic effects of the increased visceral adipose component. The muscle tissue is one of the major contributors to the peripheral action of insulin on the uptake of circulating glucose. The sarcopenic patient also has a condition of insulin resistance which can contribute to establishing and maintaining harmful metabolic circulation.

## 6. Role of Frailty and Sarcopenia in Cardiovascular Disease

Cardiovascular disease (CVD), both clinical and subclinical, has been proposed as one of the pathological conditions associated with frailty [53]. The Women’s Health Initiative Observational Study has been the first and largest study to confirm that CVD was a risk factor for developing frailty [54]. The relationship between these conditions indicates a common pathophysiological mechanism characterized by an abnormal inflammatory response, resulting in an increase in inflammatory markers, leading to chronic inflammation [55]. On the other hand, chronic inflammation is the main cause of endothelial dysfunction that leads to onset of cardiovascular diseases. The association between endothelial dysfunction and frailty confirms the role of CVD in frailty. This suggests its relevance since the early stages of vascular dysfunction (in case of only functional impairment) are apparent [56]. The endothelial dysfunction has been associated with a lot of cardiovascular risk factors: hypertension, hypercholesterolemia, diabetes mellitus, obesity, smoking. Ricci et al. [57] in a population-based study assessed that frail and pre-frail older people corresponded to a substantial proportion of those with greater CVD risk factor. In particular, diabetes mellitus (DM) seems to be the most prevalent CVD risk factor in frail and pre-frail older people. It has been demonstrated that DM is one of the strongest risk factors for atherosclerosis and, consequently, diabetic individuals present an increased risk, 3–4 times higher, of developing CVD and a double risk of mortality when compared to general population. Cacciatore et al. [58] in a 12-year survival analysis study, showed that frail individuals were 2.6 times more likely to have a complication related to DM, regarding age, sex, and number of years living with this pathology. The relationship between DM and frailty seems to be influenced by the sarcopenia. The muscle impairment in diabetic people is the result of fat infiltration in the muscle tissue, higher insulin resistance levels, the increased levels of cytokines, and reduction in motor and plates. Other important CVD risk factors in older people are hypertension and smoking, regardless of the frailty classification. No relationship has been found between frailty, obesity and waist circumference. Anyway, the concept of sarcopenic obesity behind the BMI has been introduced: frail people are characterized by sarcopenia and fat infiltration of muscle. This increased fat tissue allows the production of proinflammatory cytokines and mediators, such as interleukin-6 and C-reactive protein, which induces a state of chronic inflammation present in the frailty syndrome [59]. Therefore, it is interesting to note that aging per se induces endothelial dysfunction, in absence of cardiovascular risk factor and CVD related to increased oxidative stress and proinflammatory profile [60].

## 7. Crosstalk between Frailty and Cardiovascular Diseases in Molecular Mechanisms Level

Recent literature data suggest considering the pathophysiological mechanisms involved in the development or progression of a frailty status, for identifying frailty biomarkers [61,62]. In the context of vascular aging related to frail phenotype, several mechanisms are strongly associated with the onset of cardiovascular disease [63]. For example, inflammation is the predominant mechanism in vascular aging that induces the activation of endothelial and vascular smooth muscle cells and the migration in the wall of leukocytes. They may evolve in atherosclerosis condition or in other degenerative pathological age-related conditions. According a recent review, there are 44 most important biomarkers related to frailty. They propose a core panel of 19 high priority markers and an expanded panel with 22 medium priority markers [64]. In addition, three low priority markers are reported. These markers might be assembled in different groups according the mechanism in which they are involved: inflammation, mitochondria and apoptosis, calcium homeostasis, fibrosis, neuromuscular junction and neurons, cytoskeleton and hormones, and others. Alterations in immune system seem to be one of the most important triggers related to vascular aging. According Monti et al. [65], the aging process is related to a systemic increase in proinflammatory mediators from various sources. In addition, aging induces important changes in immune cell phenotypes and function, called “immunosenescence”. It is characterized by a shift from lymphoid to myeloid differentiation was described for B and T cell populations. Equally, there is a change in the function and receptor signaling (i.e., Toll-like receptors and RAGE) in monocytes, macrophages, dentritic cells, and neutrophils. Moreover, immune cells go through “immunosenescence” process [66]. These cells change their surface marker expression, reduce the production of reactive oxygen species (ROS) and their migration capacity, increase the production of proinflammatory over anti-inflammatory cytokines. All these events induce the release of inflammatory molecules that might be used as vascular aging and fragility biomarkers. The most important inflammatory markers are CD14 antigen also known as myeloid cell-specific leucine rich glycoprotein [67]; CX3CL1 (C-X3-C motif chemokine ligand 1, aka fractalkine) [68,69]; pentraxin [70,71]; sVCAM (soluble vascular cell adhesion molecule 1/soluble intercellular adhesion molecule 1) [72,73]; IL-6 (interleukin 6) [74,75]; CXCL 10 (C-X-C motif chemokine 10) [76]; defensins (a large family of antimicrobial and cytotoxic peptides involved in host defense and in immunomodulation) [77]. Among these seven potential inflammatory biomarkers for frailty, three seem to have high priority (IL-6, CXCL10, CX3CL1), three medium priority (pentraxin, sVCAM/sICAM, defensin), and one low priority candidate (CD14) [64]. The other group of frailty biomarkers is related to the impairment of mitochondrial function and apoptosis typical of several ageing disorders [78]. Among these the most important are GDF15 (growth differentiation factor 15 or myomitokine) [79]; FNDC5 (fibronectin type III domain containing 5) [80]; Vimentin (type III intermediate filament protein) [81]; APP (amyloid precursor protein beta) [82]; LDH (lactate dehydrogenase) [83]. From the five markers in the mitochondria and apoptosis category, the profile of GDF15, FNDC5 and vimentin are considered high priority biomarkers [64]. In addition, because in aged people the calcium homeostasis is usually altered, changes in calcium signaling and/or binding proteins have been proven to be effective markers of cellular and tissue dysfunction in these patients. The three “calcium homeostasis” biomarkers of fragility are S100B (S 100 calcium binding protein B) [84]; SMP30 (senescence-marker protein 30) [85]; calreticulin (a multifunctional protein initially identified as a Ca^2+^ storage protein) [86]. Both regucalcin and calreticum reached high priority. Another important group of biomarkers is related to fibrotic changes that various tissues show with age. One of the most important factors involved in the fibrosis is the transforming growth factor-ß (TGF-ß) [87]. A higher concentration of TGF-ß seems to be related to various diseases associated to age and fragility such as atherosclerosis, acute and chronic liver and kidney disease, autoimmunity, osteoarthritis, and neurodegenerative diseases [88]. Other fibrosis markers related to the TGF-ß pathway activation are PAI-19 (plasminogen activator inhibitor1) [89]; PLAU (urokinase plasminogen activator) [90]; MMP7 (matrix metalloproteinases 7) [91]; TGM2 (transglutaminase 2) [92]; THBS2 (thrombospondin 2) [93]; AGT100 (angiotensinogen) [94]. Yet, with the aging syndrome there is a damage of the cell cytoskeleton that leads to hormones dysregulation [95]. Consequently, a lot of hormones could likely be used as frailty biomarkers, such as the GH (growth hormone), IGF (insulin-like growth factor-1), FGF23 (fibroblast growth factor 23), resistin, adiponectine, leptin, and ghrelin.

## 8. Clinical Impact of Frailty and Sarcopenia

Frailty is emerging as a new and more specific predictor of morbidity and mortality in patients with CVD [96] (acute myocardial infarction, heart failure, heart valve disease). On the other hand, frailty seems to be a more accurate perioperative risk score than those currently used in patients underwent cardiac surgery and transcatheter aortic valve replacement. Recently, Graham et al. [97] analyzed the prognostic significance of frailty measure by the EFS in patients ≥65 years of age admitted to the hospital with acute coronary syndrome. Patients with higher EFS scores (i.e., more frailty) were older, had greater comorbidity, and were less suitable for revascularization. An EFS ≥7 was related to a longer duration of hospitalization and mortality compared with those with an EFS score ≤3. In a cohort of older patients hospitalized with non-ST-elevation myocardial infarction, Ekerstand et al. [98] assessed that frailty, quantified by the clinical frailty score (CFS), was an independent predictor of major adverse events (death, reinfarction, revascularization, major bleeding, stroke or renal replacement therapy, rehospitalization) at 1 month. In a follow up, Sanchis et al. [99] analyzed the relationship between clinical factors and laboratory parameters (i.e., inflammation, coagulation activation, hormonal dysregulation, nutritional status, kidney and cardiac function), frailty and a composite of death/myocardial infarction among survivors of ACS at the time of hospital discharge. Four clinical variables (age ≥75 years, female sex, ischemic heart disease, heart failure) and three laboratory variables (hemoglobin ≤125 g/L, vitamin D level ≤9 mg/dL, cystatin Clevel ≥1.2 mg/dL) had a predictor power similar to that of the Fried criteria for the composite outcome. Frailty might also have important prognostic implications also in patients with heart failure (HF). About 18–54% of patients with heart failure (HF) are frail [100]. A baseline frail state was found to independently predict incident HF [101]. Frailty was associated with higher likelihood of hospitalization for HF decompensation and 1-year mortality [102]. On the other hand, the presence of frailty at the time of left ventricular assist device implantation in patients with end stage-HF was shown to be associated with longer recovery time, a risk for rehospitalization and mortality [103]. Frailty has been increasingly recognized to have significance in predicting the risk of perioperative complications, resource use, and outcomes after cardiac surgery. In a lot of studies, frailty seems to improve and outperform conventional perioperative risk scores for predicting adverse outcomes [104]. A recent systematic review showed a significant association with postoperative mortality and major cardiac and cerebrovascular events (MACCE) and frailty [105]. In addition, Sudermann et al. [35] assessed a moderate correlation of the frailty with the EuroSCORE and the Society of Thoracic Surgeons (STS) scores to predict mortality in patients aged ≥74 years who were referred to cardiac surgery. In conditions where new scale is used (SHARE-FI scale), frailty seems to be better than the EuroSCORE II in predicting 1-year mortality. The role of frailty evaluation in predicting morbidity and mortality after cardiac surgery has been studied in different types of surgical procedures: aortic valve replacement [106], mitral valve surgery [107], coronary artery bypass surgery [108]. Finally, the most important application of the frailty evaluation in terms of prognostic factor is the transcatheter aortic valve replacement (TAVR) field [109]. In fact, patients undergoing TAVR are generally older, have multimorbidities and are frail. Stortecky et al. evaluated the association between preoperative multidimensional geriatric assessment (MGA) and 30-day and 1-year risk of MACCE and mortality among 100 patients undergoing TAVR. Nearly all domains of the MGA evaluated showed association with MACCE and death [110].

## 9. Dietary Intake to Prevent Sarcopenia in Patients Undergoing Cardiac Surgery

The cardiac operation is a moment of stress for ill patients. Metabolic demands and muscle breakdown are accelerated by bedrest and poor oral intake causing an important loss of muscle mass. This is particularly deleterious for frail older patients, who have lower reserves of muscle mass and strength [111]. Frailty increases the age-related changes in protein and muscle metabolism by increasing the rate of protein catabolism and decreasing the response to anabolic factors. A correct protein intake is necessary in particular in ill patients [112] (Table 1). The American Society for Parenteral and Enteral Nutrition (ASPEN) advised a protein intake of 2.0 g per kilogram of body weight per day (g/kg/d) [112]. Instead the European Society for Parenteral and Enteral Nutrition (ESPEN) recommends 1.5 g/kg/d [113]. On the other hand, these guidelines recommended aggressive postoperative nutritional support, including early enteral nutrition when necessary to meet the postoperative caloric and protein needs. In fact, in the 6 weeks after cardiac surgery, older adults lose on average 5% of their body mass, and this increased the risk of readmission in the hospital [114]. A study of patients admitted to the surgical intensive care unit showed that those with a postoperative protein deficit were less likely to be discharged home [115]. A multinational study revealed that consuming close to the recommended protein intake was associated with 60-day survival and ventilator free days [116]. A prospective interventional study demonstrated that aggressive protein supplementation was associated with a 66% reduction in infectious complications in the surgical intensive care unit [117]. A retrospective study of 1007 postsurgical patients at eight hospitals found that those with sufficient protein intake, defined as >60% of the recommended protein intake, had decreased length of stay and hospital costs [118]. The Nutrition Care in Canadian Hospitals (NCCH) study showed that surgical patients who ate less than half of the provided food had signs of malnutrition and increased length of stay [119]. To improve the nutrition of patient candidates for surgery, one of the core components of the ERAS program is a recommendation to liberally prescribe oral nutritional supplements in the pre- and postoperative periods [120]. Exercise plays a key role in the prevention and treatment of sarcopenia and today it is the most effective approach. Through the stimulus given by physical activity, numerous pathways are activated at the muscle level that converge towards anabolic pathways, with positive consequences on trophism and muscle quality. In particular, it is the moderate intensity resistance exercises that produce the most results. Intense exercise does not bring further benefits, if not actually harmful [121].

## 10. Conclusions

Frailty is the major expression of a decreased resistance to stressors, and consequently an increased vulnerability to additional diseases in elderly people. Sarcopenia can be considered a biological substrate of physical frailty. The vascular aging related to frail phenotype reflects the high susceptibility for cardiovascular diseases and negative postoperative outcomes after cardiac surgery. For this reason, a frail phenotype is a risk factor of mortality and morbidity for several diseases. A lot of biomarkers have been identified as expression of the crosstalk between frailty and cardiovascular diseases in molecular mechanisms level.

Malnutrition plays a key role in the pathogenesis of sarcopenia and frailty. Malnutrition is defined as an imbalance condition of protein or other nutrient imbalance, responsible for negative effects on body composition, physical function and clinical outcome. An optimal nutrition status and protein intake, as well as a moderate intensity resistance exercise are the key points. Accordingly, in the future the role of a specialized team-workers will be very important in the management of cardiac surgery patients. Cardiac surgeons, cardiologists, geriatricians, physiatrists, and dieticians should work in a complementary way in order to better prepare patients undergoing cardiac surgery and reduce postoperative mortality and morbidity.

## Figures and Tables

**Table 1 nutrients-13-00517-t001:** Dietary protein intake and sarcopenia.

Study Type	Duration of the Study	Material and Methods	References	Main Findings
Review	The articles were conducted from January 2010 until April 2015	The first group, containing eight articles, discussed protein or amino acid supplementation alone on sarcopenia. The second group, containing six articles, discussed exercise alone on sarcopenia. The last group, containing six articles, discussed both protein or amino acid supplementation and exercise on sarcopenia	Naseeb MA et al. [122]	Protein intakes should exceed the current recommended dietary allowance RDA (0.8 g/kg body mass per day)
The European Society for Clinical Nutrition and Metabolism (ESPEN) hosted a Workshop on Protein Requirements in the Elderly		Healthy older people for older people who are malnourished or at risk of malnutrition because they have acute or chronic illness. Older adults above 65 years	Deutz NE et al. [121]	Higher protein intakes in older adults in relation to the current protein RDA: a 25–50% increase for healthy individuals, a 50–90% increase those suffering from acute or chronic disease, and greater than 50% increase above the RDA for those experiencing severe illness or injury
Observational and cross-sectional study	Noninstitutionalized participants from the 2005-2014 National Health and Nutrition Examination Survey	Data from 11,680 adults were categorized into 51–60 years (*n* = 4016), 61–70 years (*n* = 3854), and 71 years and older (*n* = 3810) for analysis	Krok-Schoen JL et al. [123]	Over 30% failed to meet the current protein RDA
Parallel-group randomized trial, protein consumption at the current RDA or twice the RDA (2RDA) affects skeletal muscle mass and physical function in elderly men	Before treatment and after 10 wk of intervention	29 men aged > 70 y (mean ± SD) body mass index (in kg/m^2^): 28.3 ± 4.2	Mitchell CJ et al. [124]	Increasing protein intake to twice RDA (1.6 g/kg per day) resulted in significant gains in lean tissue mass in healthy older men
The current evidence related to dietary protein intake and muscle health in elderly adults	/	Elderly population of the United State	Baum JI et al. [125]	The consumption of dietary protein consistent with the upper end of the AMDRs (as much as 30–35% of total caloric intake) may prove to be beneficial, although practical limitations may make this level of dietary protein intake difficult The consumption of high-quality proteins that are easily digestible and contain a high proportion of EAAs lessens the urgency of consuming diets with an extremely high protein content
A meta-analysis of randomized controlled trials to investigate effect of whey protein supplementation on long- and short-term appetite	/	Eight publications met inclusion criteria, five records were on short-term and three records on long-term appetite	Mollahosseini M et al. [126]	Increasing daily protein intake to twice the RDA translates to an 80 kg older adult consuming about 130 protein daily. Given that protein increases satiety in a dose-dependent manner
Review		Seniors over 50 with reduced protein intake	Paddon-Jones D et al. [127]	Results from muscle protein anabolism, appetite regulation and satiety research support the contention that meeting a protein threshold (approximately 30 g/meal) represents a promising strategy for middle-aged and older adults concerned with maintaining muscle mass while controlling body fat
A multicenter, randomized, double-blinded, controlled trial with evaluation the effects of two high-quality oral nutritional supplements (ONS) differing in amount and type of key nutrients in older adult men and women	A 24-week intervention period with two energy-rich (330 kcal) ONS treatment groups	Malnourished and sarcopenic men and women, 65 years and older (*n* = 330)	Cramer JT et al. [128]	The recommendation to increase protein intake while simultaneously maintaining, and in many cases increasing, energy intake can present a protein paradox. Dietary supplementation strategies to increase protein intake may unintentionally result in partial energy redistribution, which may negatively affect both protein and energy intake
Experimental protocol that compared the stimulatory role of leucine, BCAA and EAA ingestion on anabolic signaling following exercise	/	Eight healthy male volunteers with mean (± E) age 27 ± 2 yr, body weight 84 ± 3 kg, height 181 ± 3 cm, and maximal leg strength 430± 13 kg	Moberg M et al. [129]	The capacity of resistance exercise to sensitize muscle to the anabolic potential of dietary protein is primarily achieved through a timely supply of EAA
All trials were single-blind, randomized, and counterbalanced	All laboratory visits were separated by a minimum of 7 days	Seven (*n* = 7) younger (18–45 years; four males, three females) and seven (*n* = 7) older (60–80 years; four males, three females) volunteers	Lees MJ et al. [130]	The ingestion of a novel, gel-based, leucine-enriched EAA supplement results in substantial aminoacidemia and anabolic signaling in younger and older individuals. This formulation can augment dietary protein consumption, intracellular anabolic signaling, and aminoacidemia in older adults without deleterious effects on appetite and subsequent energy intake
A systematic review of interventional evidence was performed through the use of a random-effects meta-analysis model	The effect of dietary protein supplementation during prolonged (>6 wk) resistance-type exercise training	680 subjects	Cermak NM et al. [131]	Gains in lean tissue mass were of greater magnitude in both younger and older adults when combining resistance training with protein supplementation vs. resistance training alone. Increases in type II muscle fiber cross-sectional area in older adults following resistance training
A systematic review, meta-analysis and meta-regression	Only randomized controlled trials with RET ≥6 weeks in duration and dietary protein supplementation	49 studies with 1863 participants	Morton RW et al. [132]	Protein supplementation augmented muscle growth during resistance training when habitual dietary protein intakes were, on average, below 1.6 g/kg body mass per day in younger adults. However, the impact of protein supplementation on muscle mass was reduced with advancing age
Available studies linking protein intake with physical function and health parameters in elderly 80 years old or older	/	Elderly cohorts including very old participants aged 80 years and older	Franzke B et al. [133]	The amino acid composition of a given protein source can influence the extent and amplitude of postprandiam MPS, and induce varying patterns of aminoacidemia
Studies assessing the relation between dietary protein intake and indexes of muscle mass, physical function, distribution, and muscle mass and function	/	Persons aged > 80 y sarcopenic	Traylor DA et al. [134]	The leucine content of a given protein source is particularly important in attenuating declines in a muscle mass when consumed alongside other essential amino acid EAA
Clinical trials anabolic response to essential amino acid plus whey protein composition is greater than whey protein alone in young healthy adults	/	16 healthy male and females. Characteristics: age, body weight, body mass index, lean body mass, fat mass	Park S et al. [135]	Provision of ample dietary EAA and leucine are necessary to support a skeletal muscle anabolic response in older adults. Nutritional supplementation with EAA and leucine alongside meals containing suboptimal protein content (i.e., breakfast and lunch) could assist older adults in achieving their per meal protein intake
Study time-dependent concordance and discordance between human muscle protein synthesis and mTORC1 signaling	Recruitment for healthy young men (*n* = 8; 21 ± 2 y old (mean ± SEM); body mass index (in kg/m^2^): 22.9 ± 0.9) began in January 2008	Eight postabsorptive healthy men (≈21 y of age) were studied during 8.5 h of primed continuous infusion of [1,2-^13^C_2_]leucine with intermittent quadriceps biopsies for determination of muscle protein synthesis MPS and anabolic signaling	Atherton PJ et al. [136]	When skeletal muscle is refractory to the anabolic effects of leucine during the postprandial ‘muscle-full’ period, it would be prudent that protein-based snacks or supplements are administered between meals when additional nutritional supplementation is required to reach their daily protein goal
Protocol study that demonstrates the refractoriness of muscle to nutrient-led anabolic stimulation in the postprandial period	/	Healthy, recreationally active older males (*n* = 16, 70.3 ± 2.6 years, BMI 25.5 ± 1.8 (mean ± SD) were recruited by mail and local advertising	Mitchell WK et al. [137]	Supplements may be most likely to be effective when taken in between meals, perhaps in the form of low dose EAA mixtures, rather than leucine alone; the efficacy of which may be limited in the absence of exogenous EAA to promote whole body and skeletal muscle net balance

## Data Availability

Not applicable.

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
