# Peer review of "Role of Cachexia and Fragility in the Patient Candidate for Cardiac Surgery"

_nutrients, 2021, doi:10.3390/nu13020517_

Round 1

Reviewer 1 Report

Review of “Role of cachexia and fragility in the patient candidate for cardiac surgery” (nutrients-1064177)

The topic of this study is interesting. However, several problems remain.

  1. This was narrative review, not systematic review. How to review the references were unclear.
  2. The present work is not a significant contribution, since the association between cachexia or frail and CVD, which is main topic of this study, was very small. It might be desirable to add the issue of crosstalk between fragility or sarcopenia and CVD not only in human outcome level, but also in molecular mechanisms level.

Author Response

REVIEW 1

Question: This was narrative review, not systematic review. How to review the references were unclear.

Answer: You are right. I defined the review as “narrative  review” (See pag. 2, lune 57). I rearranged and rewritten the references as per your advice. (See references)

Question:The present work is not a significant contribution, since the association between cachexia or frail and CVD, which is main topic of this study, was very small. It might be desirable to add the issue of crosstalk between fragility or sarcopenia and CVD not only in human outcome level, but also in molecular mechanisms level.

Answer: as per you advice I added a paragraph explaining the crosstalk between fragility  or sarcopenia and CVD not only in human level, but also in molecular mechanisms level. (See pag. 6-7, lines 266-328) 

Reviewer 2 Report

  1. The authors do not stress the importance of exercise already in the abstract.

  2. The authors provide definitions of "frailty" in the introduction, but none of "fragility". Further on, they seem to use the terms interchangeably. It is recommended to provide clear definitions of both terms or to adhere to "frailty" only.

  3. Line 48: "sarcopenia is considered primary", line 78 "is considered primitive": I think, "primary" is the correct term to be used.

  4. The structure of the article should be improved. For example, the considerations about the diagnosis of sarcopenia could be suitable for a particular subheading, which should precede a more detailed discussions of the clinical impact.

  5. Many statements and claims are stressed without providing an appropriate reference.

  6. The literature search should be described in detail.

  7. Extensive language editing is suggested.

Author Response

REVIEW 2

Question: The authors do not stress the importance of exercise already in the abstract.

Answer: I addedthe importance of the exercise in the abstract. (See pag 1, lines 21-23)

Question: The authors provide definitions of "frailty" in the introduction, but none of "fragility". Further on, they seem to use the terms interchangeably. It is recommended to provide clear definitions of both terms or to adhere to "frailty" only.

Answer: as per your advice I adhere to frailty only in the text.

Question: Line 48: "sarcopenia is considered primary", line 78 "is considered primitive": I think, "primary" is the correct term to be used.

Aswer: You are right. “Primary” is  the correct term that I prefer in the text to define sarcopenia age-related. (See pag. 4, lines 167-168)

Question:The structure of the article should be improved. For example, the considerations about the diagnosis of sarcopenia could be suitable for a particular subheading, which should precede a more detailed discussions of the clinical impact.

Answer: as per your adviceI added a particular subheadingfor the diagnosis of sarcopenia (See pag. 4 line 199). I discussed the clinical impact of the sarcopenia in pag 7-8, lines 330-376

Question: Many statements and claims are stressed without providing an appropriate reference.

Answer: As per your advice I provided the appropriate references for different statements.

Question: The literature search should be described in detail.

Aswer: As per your advice I described in detail the literature search (See pag. 2 lines 61-87)

Question: Extensive language editing is suggested.

Aswer: I performed a language editing.

Reviewer 3 Report

  1. Please define the process to literature searching in the abstract.
  2. There are some reviews of its kinds (i.e. Theranostics. 2019; 9, 4019-4029; J Am Coll Cardiol.2014,63,747-762; JPEN J Parenter Enteral Nutr. 2016, 40, 475-86; Nutrients. 2020 12, 3743). Please cite these and clarify how this review extends reader understanding of the topic. A clear description of the evidence gap that the review is filling is needed.
  3. The method should be expanded. The search strategy should include search terms, inclusion and exclusion criteria, databases and a period of time encompassed by the search (starting year chose to be included in this review).
  4. The paper would benefit from more comprehensive referencing- a lot of statements are made without any supporting reference (i.e. Line 68-81; Line 83-162).
  5. A lot of statements contain little synthesis of the information; i.e. how IL-1, IL-6 and TNF involved in the development of sarcopenia (Line 87-88).
  6. Define the full word when introducing an abbreviation.
  7. I miss tables that help to understand a visualize the key points (i.e. dietary intake and sarcopenia).
  8. This review comes to a weak conclusion. The authors draw conclusion from this review without future implications.
  9. Authors should follow the journal guidelines for references.

Author Response

REVIEW 3

Question: Please define the process to literature searching in the abstract.

Answer: I added  the literature searching in the abstract. (See pag 1, lines 19-20)

Question: There are some reviews of its kinds (i.e. Theranostics. 2019; 9, 4019-4029; J Am Coll Cardiol.2014,63,747-762; JPEN J Parenter Enteral Nutr. 2016, 40, 475-86; Nutrients. 2020 12, 3743). Please cite these and clarify how this review extends reader understanding of the topic. A clear description of the evidence gap that the review is filling is needed.

Answer:I added in the text the review that you advice me. (See  References 11, 18, 54, 8)

Question: The method should be expanded. The search strategy should include search terms, inclusion and exclusion criteria, databases and a period of time encompassed by the search (starting year chose to be included in this review).

Answer: I explained the method of the paper including search terms, inclusion and exclusion criteria, databases and period of time encompassed by the search (See pag. 2, lines 61-86)

Question: The paper would benefit from more comprehensive referencing a lot of statements are made without any supporting reference (i.e. Line 68-81; Line 83-162).

Answer: As per your advice I performed a more comprehensive referencing of the paper.

Question: A lot of statements contain little synthesis of the information; i.e. how IL-1, IL-6 and TNF involved in the development of sarcopenia (Line 87-88).

Answer: As per your advice I tried to better explain the molecular basis of fragility and sarcopenia. (See pag. 6-7, lines 266-328)

Question: Define the full word when introducing an abbreviation.

Answer: As per your adviceI defined the full word when introducing an abbreviation.

Question: I miss tables that help to understand a visualize the key points (i.e. dietary intake and sarcopenia).

Answer: As per your advices I introduced Table 1

Question:This review comes to a weak conclusion. The authors draw conclusion from this review without future implications.

Answer: I performed a more comprehensive conclusions. I also introduced futire implications. (See pag. 9, lines 416-440)

Question: Authors should follow the journal guidelines for references.

Answer: I rewrite the references following the journal guidelines.

Round 2

Reviewer 1 Report

No further comments.

Author Response

Many Thanks

Reviewer 2 Report

The authors claim that they strictly adhere to the term "frailty" instead of "fragility" throughout the manuscript. In fact, "fragility" is not only found in the title, but also several times in the text.

Therefore, once again, the authors should either define the difference between frailty and fragility, or, if there is none, should only use "frailty". 

Author Response

You are correct, I modified in the text the word "fragility" with "frailty".

Reviewer 3 Report

Dear Authors,

I believe that the paper has been greatly improved by these revisions. I have only one comment- Table 1: needs to be clarified. It is not clearly organized. For example, the table could have more than two columns such as: Group/study size, Duration of study, Geographic/demographic information, main findings and reference (e.g., Naseeb et al. [number]). Please do not include the full reference in table.

Author Response

I modified the TABLE 1 as per your advices.